# Absence of High Lipoprotein(a) Levels Is an Independent Predictor of Acute Myocardial Infarction without Coronary Lesions

**DOI:** 10.3390/jcm12030960

**Published:** 2023-01-26

**Authors:** Andrea Kallmeyer, Ana María Pello Lázaro, Luis M. Blanco-Colio, Álvaro Aceña, Óscar González-Lorenzo, Nieves Tarín, Carmen Cristóbal, Carlos Gutiérrez-Landaluce, Ana Huelmos, Jairo Lumpuy-Castillo, Marta López-Castillo, Juan Manuel Montalvo, Joaquín J. Alonso Martin, Lorenzo López-Bescós, Jesús Egido, Óscar Lorenzo, José Tuñón

**Affiliations:** 1Department of Cardiology, IIS-Fundación Jiménez Díaz, 28040 Madrid, Spain; 2Faculty of Medicine, Autónoma University, 28029 Madrid, Spain; 3Laboratory of Vascular Pathology, IIS-Fundación Jiménez Díaz, 28040 Madrid, Spain; 4CIBERCV, 28029 Madrid, Spain; 5Department of Cardiology, Hospital Universitario de Móstoles, 28935 Madrid, Spain; 6Department of Cardiology, Hospital Universitario de Fuenlabrada, 28942 Madrid, Spain; 7Faculty of Medicine, Rey Juan Carlos University, 28922 Alcorcón, Madrid, Spain; 8Department of Cardiology, Hospital Universitario Fundación Alcorcón, 28922 Madrid, Spain; 9CIBERDEM, 28029 Madrid, Spain; 10Department of Cardiology, Hospital de Getafe, 28905 Madrid, Spain; 11Department of Nephrology, IIS-Fundación Jiménez Díaz, 28040 Madrid, Spain

**Keywords:** MINOCA, lipoprotein(a), interleukin-18, PCSK9, inflammation, acute myocardial infarction

## Abstract

The pathophysiological mechanisms underlying Myocardial Infarction with Non-Obstructive Coronary Artery Disease (MINOCA) are still under debate. Lipoprotein (a) [Lp(a)] has proinflammatory and prothrombotic actions and has been involved in the pathogenesis of atherosclerosis. However, no previous studies have linked Lp(a) levels with the probability of developing MINOCA. Moreover, the relationship between MINOCA and the plasma levels of other proatherogenic and proinflammatory molecules such as Interleukin-18 (IL18) and proprotein convertase subtilisin/kexin type 9 (PCSK9) has not been studied. We conducted a prospective, multicenter study involving 1042 patients with acute myocardial infarction (AMI). Seventy-six patients had no significant coronary lesions. All patients underwent plasma analysis on admission. MINOCA patients were younger (57 (47–68) vs. 61 (52–72) years; *p* = 0.010), more frequently female (44.7% vs. 21.0%; *p* < 0.001), and had lower rates of diabetes and of Lp(a) > 60 mg/dL (9.2% vs. 19.8%; *p* = 0.037) than those with coronary lesions; moreover, High Density Lipoprotein cholesterol (HDL-c) levels were higher in MINOCA patients. The absence of Lp(a) > 60 mg/dL and of diabetes were independent predictors of MINOCA, as well as female sex, high HDL-c levels, and younger age. IL-18 and PCSK9 levels were not predictors of MINOCA. During a follow-up of 5.23 (2.89, 7.37) years, the independent predictors of the primary outcome (acute ischemic events or death) in the whole sample were Lp(a) > 60 mg/dL, older age, low estimated Glomerular Filtration rate (eGFR), hypertension, previous heart failure (HF), coronary artery bypass graft, use of insulin, and no therapy with acetylsalicylic acid. In conclusion, in AMI patients, the absence of high Lp(a) levels, as well high HDL-c levels, were independent predictors of the inexistence of coronary artery disease. High Lp (a) levels were also an independent predictor of ischemic events or death.

## 1. Introduction

Lipoprotein (a) [Lp(a)] is a cholesterol-rich particle which contains a large amount of apo(a) that has been involved in the development of atherosclerosis; moreover, it has been independently associated with an increased risk of ischemic cardiovascular events [1]. Its atherogenic, as well as its prothrombotic and proinflammatory [2] properties, have been previously described in several experimental and observational studies.

Around 10–20% of the population present high levels of Lp(a) with a wide range of values [3]. As Lp(a) plasmatic concentrations are genetically determined and remain quite stable lifelong in adults, their once-in-a-lifetime quantification may become a useful tool for cardiovascular risk stratification and for optimizing prevention strategies. Appropriate cut-off levels are still uncertain; however, most studies have used 40–60 mg/dL [3], showing a good correlation with vascular events [4]. High Lp(a) concentrations have also been linked to an increased risk of aortic valve stenosis, heart failure [3], and peripheral artery disease [2].

Myocardial Infarction with Non-Obstructive Coronary Artery Disease (MINOCA) is the term used to refer to acute myocardial infarction (AMI) presenting without coronary atherosclerotic lesions. A broad spectrum of causes can lead to this entity, although the underlying patho-physiological mechanisms are still poorly understood [5]. This situation has led to a change of paradigm in the interpretation of MINOCA, not as a definite diagnosis, but as a working diagnosis process.

To date, a relationship between Lp(a) and the occurrence of MINOCA has not been reported. Moreover, the relationship between MINOCA and the plasma levels of other proatherogenic and proinflammatory molecules such as Interleukin-18 (IL18) and proprotein convertase subtilisin/kexin type 9 (PCSK9) have not been explored. In this regard, it must be stated that, although PCSK9 inhibitors do not decrease high-sensitivity C-reactive (hs-CRP) levels, there is clear evidence linking this protein with inflammation [6]. Accordingly, we studied 1042 patients with AMI in order to elucidate if the absence of high levels of Lp(a) may work as a predictor of the inexistence of coronary artery disease in this population.

## 2. Methods

### 2.1. Patients

The Biomarkers in Acute Coronary Syndrome & Biomarkers in Acute Myocardial Infarction (BACS & BAMI) studies included patients admitted to five hospitals in Madrid with either non-ST elevation acute coronary syndrome (NSTEACS) or ST-elevation myocardial infarction (STEMI). Inclusion and exclusion criteria have been defined previously [7]. NSTEACS was defined as rest angina lasting more than 20 min in the previous 24 h, or new-onset class III-IV angina, along with transient ST depression or T wave inversion in the electrocardiogram considered diagnostic by the attending cardiologist and/or troponin elevation. STEMI was defined as symptomatically compatible with angina lasting more than 20 min, with ST elevation in two adjacent leads in the electrocardiogram without response to nitroglycerin, and with troponin elevation. Exclusion criteria were age over 85 years, coexistence of other significant cardiac disorders except left ventricular hypertrophy secondary to hypertension, coexistence of any illness or toxic habits that could limit patient survival, impossibility to perform revascularization when indicated, subjects for whom follow-up was not possible, and patient refusal to participate in the study. In order to avoid variability of findings due to an excessive heterogeneity in the intervals between the acute event and blood extraction, the investigators agreed to exclude patients that were not clinically stable the sixth day after the index event.

Coronary angiograms were reviewed by at least two experienced cardiologists. MINOCA was diagnosed in the absence of obstructive coronary artery disease, defined as stenoses >70%. This cut-off point was >50% for lesions at the left main coronary artery. In addition, the cardiology team ruled out the diagnosis of myocarditis or stress cardiomyopathy.

Between July 2006 and June 2014, 2740 patients were discharged from the study hospitals with a diagnosis of NSTEACS or STEMI. Moreover, 1483 patients were excluded due to the following: age over 85 years (16.4%), presence of disorders or toxic habits limiting survival (29.8%), impossibility to perform cardiac revascularization (9.6%), coexistence of other significant cardiopathy (5.7%), impossibility to perform follow-up (16.3%), clinical instability beyond the sixth day after the index event (10.9%), refusal to participate in the study (1.5%), and impossibility of the investigators to include them (9.8%). From the 1257 patients included, 1230 completed the follow-up. For the present paper, we excluded those with unstable angina; thus, finally, we included 1042 with AMI. On admission, clinical variables were recorded, and plasma was withdrawn for analysis. Final follow-up visits were carried out in June 2016.

### 2.2. Ethics Statement

The research protocol conforms to the ethical guidelines of the 1975 Declaration of Helsinki as reflected in the a priori approval by the human research committees of the institutions participating in this study: Fundación Jiménez Díaz, Hospital Fundación Alcorcón, Hospital de Fuenlabrada, Hospital Universitario Puerta de Hierro Majadahonda, and Hospital Universitario de Móstoles. All patients signed informed consent documents. Date of approval by the Ethics Committee was 24 April 2007 (act number 05-07).

### 2.3. Study Design

At baseline, clinical variables were recorded, and twelve-hour fasting venous blood samples were withdrawn and collected in EDTA. Blood samples were centrifuged at 2500 g for 10 min and plasma was stored at −80 °C. Patients were seen every year at their hospital. At the end of follow-up, the medical records were reviewed, and patient status was confirmed through telephone contact.

The primary outcome was the combination of acute ischemic events (NSTEACS or STEMI or stroke or transient ischemic attack (TIA)) and all-cause mortality. Both NSTEACS and STEMI were defined according to guidelines [8]. In addition, a previous myocardial infarction could be diagnosed in the presence of new pathological Q waves in the electrocardiogram along with a concordant new myocardial scar identified either by echocardiography or nuclear magnetic resonance imaging [9]. Stroke was defined as the rapid onset of a neurologic defect attributable to a focal vascular territory lasting more than 24 h or confirmed by new cerebral ischemic lesions on imaging studies. TIA was defined as a transient stroke with signs and symptoms resolved within the first 24 h and without cerebral acute ischemic lesions upon using imaging techniques. Although all events were recorded for each case, patients were excluded from the Cox regression analysis after the first event. Then, although the total number of events is also described, patients that had more than one event were computed only once for these analyses.

### 2.4. Biomarker and Analytical Studies

Plasma determinations were performed at the laboratory of Vascular Pathology and at the Biochemistry Laboratory at Fundación Jiménez Díaz. The investigators who performed the laboratory studies were unaware of clinical data. Plasma levels of hs-CRP were assessed using latex-enhanced immunoturbidimetry (ADVIA 2400 Chemistry System, Siemens, Munich, Germany). Troponin was assessed using immunometric immunoassay with a mice biotin-monoclonal antibody and a luminescent reaction (Ortho Clinical Diagnostics Vitros XT 7600). Plasma Lp(a) was determined by the Binding Site reagent (Edgbaston, UK) and the SPA plus instrument (R & D Systems, Minneapolis, MN, USA). For analyses, patients were divided into those with/without Lp(a) levels > 60 mg/dL according to data from the Odyssey Outcomes Trial [10] involving more than 19,000 patients with an acute coronary syndrome (ACS), where those with Lp(a) > 60 mg/dL had an enhanced probability of developing cardiovascular events. PCSK9 was determined in duplicate using enzyme-linked immunosorbent assay (ELISA) with anti-PCSK9-sepecific antibodies (ELLA kit SPCKB-PS00321, R & D Systems, Minneapolis, MN, USA). Plasma IL-18 was determined in duplicate using ELISA with IL-18-specific antibodies (ELLA kit SPCKB-PS000501, R & D Systems, Minneapolis, MN, USA). Lipids, glucose, and creatinine determinations were performed using standard methods (ADVIA 2400 Chemistry System, Siemens, Munich, Germany). The estimated glomerular filtration rate (eGFR) was calculated using the Chronic Kidney Disease Epidemiology Collaboration equation.

### 2.5. Statistical Analysis

Quantitative data following a normal distribution are presented as mean ± standard deviation (SD), and those with an abnormal distribution are displayed as median (interquartile range). Qualitative variables are presented as percentages.

Differences in baseline data of patients with and without coronary lesions were assessed using chi-squared or Fisher exact test for qualitative data. For quantitative variables, a t-Student test was performed for those following a normal distribution, and a Mann–Whitney test was used in those not normally distributed. Binary logistic regression was used to assess independent predictors of MINOCA.

Univariable Cox regression was performed to analyze which variables were associated with the development of the outcome. Then, multivariable regression analysis was carried out, including those variables that achieved statistical significance in the univariable analyses.

Analyses were performed with SPSS 19.0 (SPSS Inc., New York, NY, USA) and were considered significant when “*p*” was lower than 0.05 (two-tailed).

## 3. Results

### 3.1. Baseline Characteristics of the Population

Patients with non-obstructive coronary lesions were more frequently female and younger, concordant with previous studies [11] (Table 1). Moreover, they had a lower percentage of diabetes; however, there were no other significant differences in the rest of the traditional cardiovascular risk factors and previous medical history, including drug therapy.

HDL-c levels and eGFR were higher in patients with MINOCA. Regarding levels of markers related to inflammation, hs-CRP levels were lower in the MINOCA subgroup with a lower prevalence of Lp(a) levels > 60 mg/dL than in patients with coronary lesions. There were no differences in IL-18 and PCSK9 plasma levels. Patients without coronary lesions presented more often as non-STEMI, with lower levels of troponin and a lower presence of ejection fraction < 40% after the index event.

### 3.2. Predictors of the Absence of Coronary Lesions

Relating to factors influencing the presence of MINOCA, binary logistic regression analysis of the absence of coronary lesions (Table 2) showed that the female sex, without high Lp(a) levels and diabetes, were associated with the diagnosis of MINOCA. HDL-c plasma levels were directly associated with this diagnosis, while age and hs-CRP showed an inverse association with the existence of MINOCA.

Stepwise logistic regression analysis (Table 3) confirmed the absence of high Lp(a) levels and diabetes as independent predictors of MINOCA, along with female sex. Young age and high HDL plasma levels were also independent predictors of this diagnosis.

### 3.3. Predictors of Cardiovascular Events during Follow-Up

Patients were followed over 5.23 (2.89, 7.37) years. A total of 242 events were recorded during the follow-up, and 210 patients presented with either an ischemic event or death. Among the total number of events, 105 were ACS, 43 were stroke/TIA, and 94 were deaths. Thirty-two patients had more than one event (6 ACS and stroke/TIA, 17 ACS and death, and 9 stroke/TIA and death).

Regarding the causes of death, 38 had a cardiovascular origin, 18 were related to cancer, 12 to infections, 2 to pancreatitis, 1 to digestive bleeding, 9 to other causes, and 12 to unknown causes. Appendix A shows the results of univariable Cox hazard model analysis for the main outcome of acute ischemic events and death. The absence of coronary lesions showed a trend towards a lower incidence of the primary outcome that did not reach statistical significance. Variables displaying “*p*” values < 0.05 were included in the multivariable analysis with the stepwise Cox hazard models (Table 4). High Lp(a) levels were an independent predictor of adverse outcomes along with hypertension, previous heart failure and coronary artery bypass graft, the prescription of insulin at discharge from the index event, and older age. The use of aspirin on discharge and eGFR showed an inverse and independent association with the incidence of the primary outcome.

## 4. Discussion

### 4.1. MINOCA Prevalence, Pathophysiology and Diagnostic Criteria

MINOCA accounts for 5–15% of ACS [12], and, despite the fact it usually confers better prognosis than ACS with significant coronary artery disease, it has been highlighted recently as a non-trivial condition [11,13]. Three diagnostic criteria have been established for this entity: evidence of myocardial infarction according to clinical practice guidelines [14], significant coronary artery disease ruled out by coronary angiogram, and no clinically overt acute condition that may explain troponin rise [15]. Ischemia is the most frequent cause of MINOCA, and this can be due to plaque rupture or erosion, spontaneous coronary dissection, oxygen supply–demand imbalance, vasospasm, and cardiotoxicity [16]. It is crucial to remark that the absence of obstructions on the epicardial arteries does not mean the inexistence of endothelial damage, mild to moderate atherosclerosis lesions, or microvascular disease [16].

### 4.2. Inflammation and MINOCA

Inflammation plays a key role in the pathogenesis of atherothrombosis. Some papers have suggested that a relation between MINOCA and inflammation also exists, showing enhanced pericoronary fat attenuation [17] and even increased levels of some inflammatory biomarkers such as pappalysin-1 (PAPPA), indicating a higher degree of inflammation in this entity than in MI-CAD [18]. However, the amount of information relating to MINOCA and inflammation is still scarce. This is one of the reasons why MINOCA emerges as a field of opportunity for investigation, one which is especially oriented to the identification of prognostic and/or predictor factors, as its influence on outcomes is not unremarkable.

### 4.3. Relationship between Lp(a) and MINOCA

In our cohort, we have analyzed for the first time the potential relationship of Lp(a) levels with the absence of coronary lesions in patients with AMI. We observed that MINOCA patients had high Lp(a) levels less often and showed lower hs-CRP levels than those with coronary lesions; however, there were no differences in IL-18 and PCSK9 plasma levels. In addition, MINOCA patients were more commonly female, younger, less frequently diabetic, and had better eGFR and higher levels of HDL-c than patients with coronary lesions [15]. These findings are relevant since Lp(a) has proinflammatory, pro-oxidative, and prothrombotic actions and has been involved in the genesis of atherosclerosis [1,2,19].

### 4.4. Predictors

At stepwise logistic regression analysis, the absence of both, high Lp(a) levels and diabetes, high HDL-c levels, younger age and female sex were independent predictors of MINOCA. These data strongly suggest that atherosclerosis-prone factors, as high Lp(a), and the traditional cardiovascular risk factors, may have a lesser contribution to the development of MINOCA. These findings point to probable differences in the pathogenesis of MINOCA as compared to MI-CAD. In this regard, although Hs-CRP levels did not reach the statistical significance as an independent predictor of MINOCA, at univariable binary logistic regression, they were inversely associated with this diagnosis. Then, it may be hypothesized that the pathophysiological mechanisms of MINOCA and AMI with coronary lesions are not identical, as it seems that a high inflammatory status and cardiovascular risk burden is related to the existence of significant atherosclerosis.

### 4.5. Lp(a) as an Outcome Predictor after an AMI

The absence of coronary lesions showed a trend towards a lower incidence of ischemic events and death during the follow-up that probably did not reach statistical significance due to the limited number of patients with MINOCA. However, Lp(a) levels higher than 60 mg/dL were independent predictors of this outcome, suggesting that high plasma levels of this molecule may have a role in predicting recurrent ischemic events, as has been shown in other studies [20]. Moreover, hypertension, previous heart failure, coronary artery bypass graft, insulin therapy, and older age were independently related to the development of the outcome, while treatment with acetylsalicylic and eGFR were inversely and independently associated with it. Then, a high burden of cardiovascular risk factors along with markers of myocardial and extensive vascular damage was associated with a worse prognosis.

### 4.6. Future Directions of Research in this Field

Although these results do not change the current management of ACS, they emphasize the relevance of Lp(a) in the development of coronary artery disease and the potential differences in the pathophysiology of MINOCA as compared with AMI with obstructive coronary disease. In this regard, it will be of interest to know if the potential benefit of Lp(a) lowering, which is being tested in some clinical trials [21], translates into an exclusive reduction in ACS with obstructive coronary disease without affecting the development of MINOCA. In addition, the potential usefulness of high Lp(a) levels as a prognostic biomarker in AMI patients should be considered in clinical practice if these results are consistently reproduced by other investigators. Finally, in case Lp(a) lowering is demonstrated to be of benefit in the future, this could influence the election of lipid-lowering therapy in patients with high Lp(a) levels.

### 4.7. Limitations of the Study

This study has some limitations. First, our findings could not be applicable to a series of AMI patients with severe left ventricular dysfunction, as the percentage of cases with ejection fraction <40% in our population was low. This was due to the design of the study, as patients should undergo plasma extraction in a stable condition no later than six days after admission. Patients who are unstable by that time probably had left ventricular dysfunction more often. However, extracting plasma in this situation would have led to confounding results. Moreover, awaiting until the recovery of these unstable patients for blood extraction would have implied a great delay and a lack of homogeneity in the timing of blood sample collection. Finally, although the size of the whole population was quite large, the number of patients with a diagnosis of MINOCA was limited, and this may have resulted in insufficient statistical power to detect more differences between the two types of patients.

## 5. Conclusions

Among patients with AMI, the absence of high Lp(a) appears to be an independent predictor of the inexistence of coronary lesions. This finding may help to gain a better understanding of MINOCA settings and a more accurate prediction of ischemic events in this population.

## Figures and Tables

**Table 1 jcm-12-00960-t001:** Baseline characteristics of the population according to the presence of coronary lesions.

Parameter		No Coronary Lesion (*N* = 76)	Coronary Lesions (*N* = 966)	*p*	
Sex (Male) (*n*, %)		42 (55.3%)	763 (79.0%)	<0.001	
Age (years)		57 (48–68)	61 (52–72)	0.010	
Race (Caucasian) (*n*, %)		72 (94.7%)	937 (97.0%)	0.497	
Estimated Glomerular Filtration Rate (mL/min/1.73 m^2^)		86 (74–98)	82 (66–94)	0.032	
Smoker (*n*, %)		33 (43.4%)	443 (45.9%)	0.681	
Diabetes (*n*, %)		7 (9.2%)	216 (22.4%)	0.007	
Hypertension (*n*, %)		43 (56.6%)	538 (55.7%)	0.881	
Dyslipidemia (*n*, %)		40 (52.6%)	561 (58.1%)	0.355	
Previous Myocardial Infarction (*n*, %)		4 (5.3%)	113 (11.7%)	0.087	
Peripheral Artery Disease (*n*, %)		5 (6.6%)	46 (4.8%)	0.480	
Cerebrovascular Accident (*n*, %)		1 (1.3%)	33 (3.4%)	0.321	
Atrial Fibrillation (*n*, %)		2 (2.6%)	21 (2.2%)	0.794	
Previous Heart Failure (*n*, %)		0 (0.0%)	9 (0.9%)	0.398	
Previous Coronary artery bypass graft (*n*, %)		0 (0.0%)	36 (3.7%)	0.087	
Body Mass Index (kg/m^2^)		27 (25–30)	28 (25–31)	0.475	
Familiar background (*n*, %)		18 (23.7%)	238 (24.6%)	0.385	
Eating Fruit (servings per day)		2.5 (1.0–3.9)	2.5 (1.2–3.2)	0.812	
Eating Fish (per week)		2.0 (1.6–3.0)	2.0 (1.5–3.0)	0.699	
Alcohol consumption	0	28 (37.3%)	385 (40.9%)	0.200	
	1–7	33 (44.0%)	293 (31.1%)		
	8–14	5 (6.7%)	140 (14.9%)		
	>14	9 (12.0%)	121 (12.9%)		
Index Cardiovascular Event	NSTEMI	55 (72.4%)	379 (39.2%)	<0.001	
	STEMI	21 (27.6%)	587 (60.8%)		
Number of Vessels Diseased (*n*)		0 (0–0)	1 (1–2)	<0.001	
Ejection fraction < 40 (*n*, %)		4 (5.3%)	161 (16.7%)	0.028	
Type of revascularization	No	77 (100%)	69 (7.1%)	<0.001	
	Covered Stent	0 (0.0%)	541 (56.0%)		
	Standard Stent	0 (0.0%)	282 (29.2%)		
	Angioplasty	0 (0.0%)	34 (3.5%)		
	Surgical revascularization	0 (0.0%)	40 (4.1%)		
Troponin I (ng/L)		3.9 (1.0–18.1)	19.7 (4.0–70.0)	<0.001	
LDL-c (mg/dL)		115 ± 35	119 ± 37	0.412	
Triglycerides (mg/dL)		127 (92–174)	129 (91–180)	0.984	
HDL-c (mg/dL)		42 (35–54)	38 (32–46)	0.002	
IL-18 (ng/L)		219 (156–283)	213 (159–298)	0.899	
PCSK9 (ng/mL)		423 (338–543)	434 (353–532)	0.738	
Lp(a) > 60 mg/dL		6 (9.2%)	175 (19.8%)	0.037	
Hs-CRP (mg/L)		0.9 (0.5–2.1)	2.0 (1.0–3.9)	<0.001	
**Previous Medical Therapy**					
Acetylsalicylic acid (*n*, %)		13 (17.1%)	217 (22.5%)	0.278	
AntiP2Y12 (*n*, %)		1 (1.3%)	47 (4.9%)	0.349	
Acenocumarol (*n*, %)		1 (1.3%)	22 (2.3%)	0.583	
Statins (*n*, %)		14 (18.4%)	261 (27.0%)	0.249	
Ezetimibe (*n*, %)		1 (1.3%)	22 (2.3%)	0.583	
Fibrates (*n*, %)		0 (0%)	11 (1.1%)	0.350	
Insulin (*n*, %)		2 (2.6%)	49 (5.1%)	0.342	
Oral antidiabetic drugs (*n*, %)		5 (6.6%)	137 (14.2%)	0.162	
ACE inhibitors (*n*, %)		19 (25.0%)	310 (32.1%)	0.200	
Aldosterone receptor blockers (*n*, %)		1 (1.3%)	21 (2.2%)	0.616	
Betablockers (*n*, %)		11 (14.5%)	155 (16.0%)	0.718	
Nitrates (*n*, %)		3 (3.9%)	56 (5.8%)	0.502	
Diltiazem (*n*, %)		0 (0%)	26 (2.7%)	0.148	
Verapamil (*n*, %)		1 (1.3%)	8 (0.8%)	0.872	
Dihydropyridines (*n*, %)		3 (3.9%)	96 (9.9%)	0.086	
Diuretics (*n*, %)		9 (11.8%)	164 (17.0%)	0.247	
Proton pump inhibitors (*n*, %)		14 (18.4%)	177 (18.3%)	0.983	
Digoxin (*n*, %)		0 (0%)	6 (0.6%)	0.491	
Amiodarone (*n*, %)		0 (0%)	6 (0.6%)	0.491	

Abbreviations: ACE: Angiotensin Convertase Enzyme; HDL-c: High Density Lipoprotein cholesterol; Hs-CRP: High sensitivity C Reactive Protein; IL-18: Interleukin-18; LDL-c: Low Density Lipoprotein cholesterol; Lp(a): Lipropotein (a); NSTEMI: Non-ST Elevation Myocardial Infarction; PCSK9: Proprotein Convertase Subtilisin/Kexin type 9; P2Y12: Protein 2Y12; STEMI: ST-Elevation Myocardial Infarction.

**Table 2 jcm-12-00960-t002:** Binary logistic regression analysis of the absence of coronary lesions.

		Crude	
Parameter		OR (95% CI)	*p*
Sex (Male)		0.33 (0.20–0.53)	<0.001
Age (years)		0.97 (0.95–0.99)	0.007
Race (Caucasian)		1.86 (0.63–5.45)	0.258
Estimated Glomerular Filtration Rate		1.01 (1.00–1.03)	0.033
Smoker		0.91 (0.57–1.45)	0.681
Diabetes		0.35 (0.16–0.78)	0.010
Hypertension		1.04 (0.65–1.66)	0.881
Dyslipidemia		0.80 (0.50–1.28)	0.356
Previous Myocardial Infarction		0.42 (0.15–1.17)	0.097
Peripheral Artery Disease		1.41 (0.54–3.66)	0.482
Cerebrovascular Accident		0.34 (0.05–2.79)	0.340
Atrial Fibrillation		1.22 (0.28–5.29)	0.794
Body Mass Index		0.98 (0.93–1.04)	0.475
Familiar background		0.82 (0.46–1.47)	0.513
Eating Fruit (servings per day)		1.01 (0.90–1.14)	0.812
Eating Fish (servings per week)		1–03 (0.88–1.20)	0.699
Alcohol consumption	0	Ref.	-
	1–7	1.55 (0.91–2.62)	0.103
	8–14	0.49 (0.19–1.30)	0.151
	>14	1.02 (0.47–2.23)	0.955
LDL-c		1.00 (0.99–1.00)	0.411
Triglycerides		1.00 (1.00–1.00)	0.905
HDL-c (per 5 mg/dL increment)		1.18 (1.08–1.28)	<0.001
IL-18		1.00 (1.00–1.00)	0.900
PCSK9		1.00 (1.00–1.00)	0.490
Lp(a) > 60 mg/dL		0.41 (0.17–0.97)	0.043
Hs-CRP		0.85 (0.75–0.96)	0.009
Acetylsalicylic acid		0.71 (0.38–1.32)	0.280
AntiP2Y12		0.26 (0.03–1.91)	0.186
Acenocumarol		0.57 (0.08–4.30)	0.588
Statins		0.61 (0.33–1.11)	0.104
Ezetimibe		0.57 (0.07–4.30)	0.588
Insulin		0.51 (0.12–2.12)	0.351
Oral antidiabetic drugs		0.42 (0.17–1.07)	0.070
ACE inhibitors		0.71 (0.41–1.21)	0.202
Aldosterone receptor blockers		0.60 (0.08–4.52)	0.620
Betablockers		0.88 (0.46–1.72)	0.719
Nitrates		0.67 (0.20–2.19)	0.505
Dihydropyridines		0.37 (0.11–1.20)	0.099
Diuretics		0.66 (0.32–1.34)	0.250
Proton Pump Inhibitors		1.00 (0.55–1.84)	0.983

Abbreviations as for Table 1.

**Table 3 jcm-12-00960-t003:** Stepwise logistic regression analysis of the absence of coronary lesions.

Parameter	Odds Ratio	95% CI	*p*
Sex, male	0.34	0.19–0.60	<0.001
Age, years	0.97	0.94–0.99	0.002
Diabetes	0.40	0.17–0.97	0.043
HDL-c (per 5 mg/dL increment)	1.13	1.02–1.24	0.015
Lp(a) >60 mg/dL	0.35	0.14–0.84	0.019

Abbreviations as for Table 1.

**Table 4 jcm-12-00960-t004:** Stepwise Cox proportional hazard models for the incidence rates of acute ischemic events and death.

Parameter	Hazard Ratio	95% CI	*p*
Age, years	1.02	1.00–1.03	0.034
Estimated Glomerular Filtration Rate	0.99	0.98–0.99	0.035
Hypertension	1.50	1.06–2.13	0.021
Previous Heart Failure	1.55	1.07–2.26	0.022
Coronary Artery Bypass Grafting	2.05	1.19–3.54	0.010
Lp(a) > 60 mg/dL	1.44	1.02–2.03	0.036
Insulin	1.72	1.09–2.74	0.021
Acetylsalicylic acid	0.57	0.32–0.99	0.048

Abbreviations as for Table 1.

## Data Availability

Data is available upon request.

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
