# Peer review of "Absence of High Lipoprotein(a) Levels Is an Independent Predictor of Acute Myocardial Infarction without Coronary Lesions"

_jcm, 2023, doi:10.3390/jcm12030960_

Round 1

Reviewer 1 Report

This is an interesting study suggesting that the absence of high lipoprotein(a) levels is an independent predictor of acute myocardial infarction without coronary lesions.

However, a few comments should be noted as follows:

1.    The abbreviations should be put in brackets after their full names at the first appearance. (e.g. MINOCA and HDL-c in the abstract; MINOCA and AMI in the introduction).

2.    “The primary outcome was the combination of acute ischemic events (both, coronary and cerebrovascular) and all-cause mortality”. Please give more details about the outcome. Does it only include non-fatal AMI and non-fatal stroke, or also include unstable angina and TIA?

3.    It is more proper to use “univariable” or “multivariable” instead of “univariate” or “multivariate”.

4.    Because appropriate cut-off levels of Lp(a) are still uncertain, I suggested a sensitivity analysis using different cut-offs or regarding Lp(a) as a continuous variable or exploring the exposure-response relationship using restricted cubic splines.

5.    In the conclusion section, the authors also pointed out high HDL-c plasma levels appear to be independent predictors of the inexistence of coronary lesions. Although the authors found several predictors in the logistic and Cox regression models, I suggest the authors should focus on the results of Lp(a), both in MINOCA prediction and prognosis of AMI, to fit the title.

6.    The authors should point out the future direction of research in this field in the discussion section.

Author Response

Thank you very much for all your suggestions. We have followed them (see our answers below).

  1. Thank you very much for this observation. We have corrected it.
  2. Thank you very much for your comment. The term “acute ischemic events” includes AMI (with/without ST-elevation), unstable angina, stroke, and TIA. This has been changed in the new version of the manuscript.

  3. Thank you very much for this correction. We have proceeded to change univariate and multivariate to univariable and multivariable.

  4. We chose the cut-off level of Lp(a) > 60 mg/dL after considering this threshold as the most reliable for high cardiovascular risk profiling. This was the threshold that separated patients with the highest risk of CVE at the Odyssey outcomes clinical trial in the paper published by Bittner et al (we rounded the exact cut-off of 59.6 to 60 mg/dl). Since this trial involved more than 19,000 patients with ACS we believe that this may be a consistent cut-off. The clinical practice guidelines on dyslipidemias from the ESC only emphasize the relevance of identifying patients with extremely high Lp(a) (>180 mg/dl), but they do not provide a specifical cut-off.

    According to your suggestion, we have performed an univariable Cox regression analysis using Lp(a) as a continuous variable. As you can see below, the univariable analysis was significant. However, this statistical significance was no longer evident at multivariable analysis. We interpret that low values of Lp(a) probably lack an adequate discriminant power.

    Lp(a)

    1.01

    1.00-1.01

    0.048

    We then performed this univariable analysis using different cut-offs. As you may see below, only the cut-off level of 60 mg/dl yields a significant result, that is confirmed at multivariable analysis.

    Lp(a)>45

    1.21

    0.88-1.64

    0.228

    Lp(a)>50

    1.29

    0.95-1.77

    0.106

    Lp(a)>60

    1.60

    1.14-2.23

    0.006

          Taking this into account and, also, the fact that the chosen 60 mg/dl cut-off comes from a different (and large) population, without the existence in the literature of other relevant values unanimously accepted, we understand that keeping the performed analyses using the cut-off of 60 mg/dl may be the most consistent decision.

    5. Thank you very much for this observation. Following your recommendation, we have deleted HDL-c from the conclusions section.

    6. We sincerely appreciate this recommendation. We have included a section entitled “Future directions of research in this field” in the discussion where we give our opinion on this point. We paste it below, so you can have a look at it.

    Future directions of research in this field

    Although these results do not change the current management of ACS, they emphasize the relevance of Lp(a) in the development of coronary artery disease and the potential differences in the pathophysiology of MINOCA as compared with AMI with obstructive coronary disease. In this regard, it will be of interest to know if the potential benefit of Lp(a) lowering that is being tested in some clinical trials translates into an exclusive reduction of ACS with obstructive coronary disease without affecting the development of MINOCA. In addition, the potential usefulness of high Lp(a) levels as a prognostic biomarker in AMI patients should be considered in the clinical practice if these results are consistently reproduced by other investigators. Finally, in case Lp(a) lowering is demonstrated to be of benefit in the future, this could lead to changes in the election of lipid-lowering therapy in patients with high Lp(a) levels.

Reviewer 2 Report

The analysis concerns a significant clinical problem, which is MINOCA. A negative relationship between the Lp(a) concentration and the MINOCA rate was observed in this study. This study also confirmed the effect of increased Lp(a) levels on the higher risk of ischemic events, already found in previous studies. This analysis has rather cognitive value and it is difficult to translate its results into clinical management. The information that the level of Lp(a) concentration in patients with MINOCA is lower does not change our standard management before and after ACS, which is based on the use of lipid-lowering drugs anyway.

Author Response

Thank you very much for your reflection. We really appreciate and acknowledge it. We are aware that our findings do not translate to significant clinical practice changes concerning such a complex entity as MINOCA is. However, that was not our main purpose. On the contrary, it was our intention at the time of designing the study to help to provide an insight to the pathophysiological differences among non-coronary disease related ACS and the coronary-disease related ones. Therefore, and in line with this intellectual briefing, we interpreted our results either as an enhancement of the stablished knowledge on the field that help to reinforce that MINOCA happens due to a non-atherosclerotic mechanism, and as an additional valuable insight to highlight the usefulness of high Lp(a) levels as a prognostic marker for this entity. So much for that, we also consider that having demonstrated the latter, and taking advantage of the upcoming studies with Lp(a) lowering agents, from our results may surge in the years to come some therapeutic issue orientated to the prevention of recurrent ischemic events in patients that had had a MINOCA.

Reviewer 3 Report

Dear authors,

I have read your manuscript with great interest; however, I found several points that need to be addressed:

-) Spelling and grammar: Please let the manuscript be revised by an English native speaker. There is already a typo in the first sentence of the abstract.

-) Methods, section 2.1 (patients): It is not enough to refer entirely to another publication here. Please state your used Methods and what you mean by the described overall studies and acronyms in detail.

-) Ethics Vote: Please also state the date and number of the vote.

-) Methods, section 2.3: This reads rather unintuitive. Please give this more of a clear thread running through this section. Also, a figure with study flow diagram would help.

-) Methods, general: It must be clearly and extensively explained how you arrived at the diagnoses (MINOCA, occlusive CAD,...).

-) Discussion and Conclusion: This should be formualted a bit clearer, guiding a potential reader through these sections (maybe also through introducing subheadings) and naturally arriving at the conclusion (which should be more concise). Also, the Discussion section could benefit from more thoughts on clinical applicability of your findings.

Author Response

Thank you very much for your review and suggestions. We have followed your recommendations (see answers below).

-) Spelling and grammar: 

Thank you very much. We have already corrected that typo in the first sentence. Besides, we have sent the manuscript to the professional English-translation service of our institution. We think that this has improved considerably the quality of the manuscript.

-) Methods, section 2.1 (patients):

Thank you very much for this recommendation. We have modified the methods section to make it more straightforward and clearer.

-) Ethics vote: 

The date was the 24th of April 2007, and it was accepted by the whole Ethic Committee and documented in the act number 05-07 of that year’s exercise. We have already included this information in the body of the correspondent paragraph in the manuscript.

-) Methods, section 2.3: 

Thank you very much. As for the previous methods section 2.1, we have modified, according to your suggestion, this section 2.3 of the manuscript to clarify the patient inclusion process. We believe this may be enough, because we have shortened this section and the design is quite simple. Nevertheless, if you think that a flow chart may be of help, please let us know.

-) Methods, general: 

Thank you very much for this feedback. We have included a short paragraph in the section 2.1 of methods explaining this.

-) Discussion, conclusions: 

Thank you very much. Following your wise advice, we have included subheadings in the text body of the manuscript to improve its reading and understanding. Specifically, we have added a section entitled “Future directions of research in this field”, where we discuss the potential applicability of the results.